# MCMC Should Mix: Learning Energy-Based Model with Neural Transport Latent Space MCMC

**Erik Nijkamp**[1,3†*]
erik.nijkamp@salesforce.com

**Ruiqi Gao**[1,2*]
ruiqig@google.com

**Pavel Sountsov**[2]
siege@google.com

**Srinivas Vasudevan**[2]
srvasude@google.com

**Bo Pang**[1,3]
b.pang@salesforce.com

**Song-Chun Zhu**[1,4]
sczhu@stat.ucla.edu

**Ying Nian Wu**[1]
ywu@stat.ucla.edu

[1]Department of Statistics, UCLA   [2]Google Research   [3]Salesforce Research
[4]Beijing Institute for General Artificial Intelligence (BIGAI)

## Abstract

Learning energy-based model (EBM) requires MCMC sampling of the learned model as an inner loop of the learning algorithm. However, MCMC sampling of EBMs in high-dimensional data space is generally not mixing, because the energy function, which is usually parametrized by a deep network, is highly multi-modal in the data space. This is a serious handicap for both theory and practice of EBMs. In this paper, we propose to learn an EBM with a flow-based model (or in general a latent variable model) serving as a backbone, so that the EBM is a correction or an exponential tilting of the flow-based model. We show that the model has a particularly simple form in the space of the latent variables of the backbone model, and MCMC sampling of the EBM in the latent space mixes well and traverses modes in the data space. This enables proper sampling and learning of EBMs.

## 1 Introduction

The energy-based model (EBM) (LeCun et al., 2006; Ngiam et al., 2011; Kim & Bengio, 2016; Zhao et al., 2016; Xie et al., 2016; Gao et al., 2018; Kumar et al., 2019b; Nijkamp et al., 2019; Du & Mordatch, 2019; Finn et al., 2016; Atchadé et al., 2017; De Bortoli et al., 2021; Song & Ou, 2018) defines an unnormalized probability density function on the observed data such as images via an energy function, so that the density is proportional to the exponential of the negative energy. Taking advantage of the approximation capacity of modern deep networks such as convolutional networks (ConvNet) (LeCun et al., 1998; Krizhevsky et al., 2012), recent papers (Xie et al., 2016; Gao et al., 2018; Kumar et al., 2019b; Nijkamp et al., 2019; Du & Mordatch, 2019) parametrize the energy function by a ConvNet. The ConvNet-EBM is highly expressive and the learned EBM can produce realistic synthesized examples.

The EBM can be learned by maximum likelihood estimation (MLE), which follows an "analysis by synthesis" scheme. In the synthesis step, synthesized examples are generated by sampling from the current model. In the analysis step, the model parameters are updated based on the statistical difference between the synthesized examples and the observed examples. The synthesis step usually requires Markov chain Monte Carlo (MCMC) sampling, and gradient-based sampling such as Langevin dynamics (Langevin, 1908) or Hamiltonian Monte Carlo (HMC) (Neal, 2011) can be conveniently implemented on the current deep learning platforms where gradients can be efficiently and automatically computed by back-propagation.

---

*Equal contribution. †Majority of research was conducted at Google.

However, gradient-based MCMC sampling in the data space generally does not mix, which is a fundamental issue from a statistical perspective. The data distribution is typically highly multi-modal. To approximate such a distribution, the density or energy function of the ConvNet-EBM needs to be highly multi-modal as well. When sampling from such a multi-modal density in the data space, gradient-based MCMC tends to get trapped in local modes with little chance to traverse the modes freely, rendering the MCMC non-mixing. Without being able to generate fair examples from the model, the estimated gradient of the maximum likelihood learning can be highly biased, and the learned model parameters can be far from the unbiased estimator given by MLE. Even if we can learn the model by other means without resorting to MCMC sampling, e.g., by noise contrastive estimation (NCE) (Gutmann & Hyvärinen, 2010; Gao et al., 2019; Wang & Ou, 2018) or by amortized sampling (Kim & Bengio, 2016; Song & Ou, 2018; Grathwohl et al., 2020), it is still necessary to be able to draw fair examples from the learned model for the purpose of model checking or downstream applications based on the learned model.

Accepting the fact that MCMC sampling is not mixing, contrastive divergence (Tieleman, 2008) initializes finite step MCMC from the observed examples, so that the learned model is admittedly biased from the MLE. Du et al. (2020) improves contrastive divergence by initializing MCMC from augmented samples. Recently, Nijkamp et al. (2019) proposes to initialize short-run MCMC from a fixed noise distribution, and shows that even though the learned EBM is biased, the short-run MCMC can be considered a valid model that can generate realistic examples. This partially explains why EBM learning algorithm can synthesize high quality examples even though the MCMC does not mix. However, the problem of non-mixing MCMC remains unsolved. Without proper MCMC sampling, the theory and practice of learning EBMs is on a very shaky ground. The goal of this paper is to address the problem of MCMC mixing, which is important for proper learning of EBMs. The subpar quality of synthesis of our approach is a concern, which we believe may be addressed with recent flow architectures (Durkan et al., 2019) and jointly updating the flow model in future work. We believe that fitting EBMs properly with mixing MCMC is crucial to downstream tasks that go beyond generating high-quality samples, such as out-of-distribution detection and feature learning. We will investigate our model on those tasks in future work.

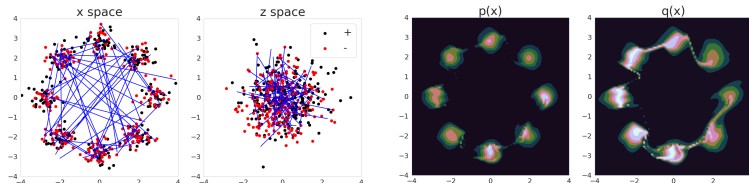

Figure 1: Demonstration of mixing MCMC with neural transport learned from a mixture of eight 2D Gaussians. The Markov chains pulled back into data space $x$ freely traverse the modes of the mixture of Gaussians. *Left*: observed examples (black) and trajectories (blue) of Markov chains (red) in data space $x$ and latent space $z$. *Right*: density estimations with exponentially tilted model $p_\theta$ and underlying flow $q_\alpha$.

We propose to learn an EBM with a flow-based model (or in general a latent variable model) as a backbone model (or base model, or core model), so that the EBM is in the form of a correction, or an exponential tilting, of the flow-based model. Flow-based models have gained popularity in generative modeling (Dinh et al., 2014; 2016; Kingma & Dhariwal, 2018; Grathwohl et al., 2018; Behrmann et al., 2018; Kumar et al., 2019a; Tran et al., 2019) and variational inference (Kingma & Welling, 2013; Rezende & Mohamed, 2015; Kingma et al., 2016; Kingma & Welling, 2014; Khemakhem et al., 2019). Similar to the generator model (Kingma & Welling, 2013; Goodfellow et al., 2014), a flow-based model is based on a mapping from the latent space to the data space. However, unlike the generator model, the mapping in the flow-based model is deterministic and one-one, with closed-form inversion and Jacobian that can be efficiently computed. This leads to an explicit normalized density via *change of variable*. However, to ensure tractable inversion and Jacobian, the mapping in the flow-based model has to be a composition of a sequence of simple transformations of highly constrained forms. In order to approximate a complex distribution, it is necessary to compose a large number of such transformations. In our work, we propose to learn the EBM by correcting a relatively simple flow-based model with a relatively simple energy function parametrized by a free-form ConvNet.

We show that the resulting EBM has a particularly simple form in the space of the latent variables. MCMC sampling of the EBM in the latent space, which is a simple special case of neural transport MCMC (Hoffman et al., 2019), mixes well and is able to traverse modes in the data space. This enables proper sampling and learning of EBMs. Our experiments demonstrate the efficacy of learning EBM with flow-based backbone, and the neural transport sampling of the learned EBM greatly mitigates the non-mixing problem of MCMC.

## 2 RELATED WORK AND CONTRIBUTIONS

The following are research themes in generative modeling and MCMC sampling that are closely related to our work.

**Neural transport MCMC**. Our work is inspired by neural transport sampling (Hoffman et al., 2019). For an unnormalized target distribution, the neural transport sampler trains a flow-based model as a variational approximation to the target distribution, and then samples the target distribution in the space of latent variables of the flow-based model via change of variable. In the latent space, the target distribution is close to the prior distribution of the latent variables of the flow-based model, which is usually a unimodal Gaussian white noise distribution. Consequently the target distribution in the latent space is close to be unimodal and is much more conducive to mixing and fast convergence of MCMC than sampling in the original space (Mangoubi & Smith, 2017).

Our work is a simplified special case of this idea, where we learn the EBM as a correction of a pre-trained flow-based model, so that we do not need to train a separate flow-based approximation to the EBM. The energy function, which is a correction of the flow-based model, does not need to reproduce the content of the flow-based model, and thus can be kept relatively simple. Moreover, in the latent space, the resulting EBM takes on a very simple form where the inversion and Jacobian in the flow-based model disappear. This may allow for using free-form flow-based models where inversion and Jacobian do not need to be in closed form (Grathwohl et al., 2018; Behrmann et al., 2018), or more general latent variable models.

**Energy-based corrections**. Our model is based on an energy-based correction or an exponential tilting of a more tractable model. This idea has been explored in noise contrastive estimation (NCE) (Gutmann & Hyvärinen, 2010; Gao et al., 2019) and introspective neural networks (INN) (Tu, 2007; Jin et al., 2017; Lazarow et al., 2017), where the correction is obtained by discriminative learning. Earlier works include Rosenfeld et al. (2001); Wang & Ou (2018). Recently Xiao et al. (2020) recruits an EBM to correct a variational autoencoder with MCMC-based learning methods. Correcting or refining a simpler and more tractable backbone model can be much easier than learning an EBM from scratch, because the EBM does not need to reproduce the knowledge learned by the backbone model. It also allows easier sampling of EBMs.

**Amortized sampling**. Non-mixing MCMC sampling of an EBM is a clear call for latent variables to represent multiple modes of the original model distribution via explicit top-down mapping, so that the distribution of the latent variables is less multi-modal. Earlier works in this direction include Bengio et al. (2013); Kim & Bengio (2016); Dai et al. (2017); Song & Ou (2018); Brock et al. (2018); Xie et al. (2018); Han et al. (2019); Kumar et al. (2019b); Grathwohl et al. (2020). In this paper, we choose to use flow-based model for its simplicity, because the distribution in the data space can be translated into the distribution in the latent space by a simple change of variable, without requiring integrating out extra dimensions as in the generator model.

**Proper learning of EBMs**. Wang & Ou (2017) studies the proper learning of EBMs in the modality of languages and recruits Gibbs sampling from the discrete distributions. In comparison, our work concerns images in continuous space for which we sample by gradient-based MCMC. Moreover, our work emphasizes the empirical evaluation of the mixing behavior of Markov chains.

**Contributions**. This paper tackles the problem of non-mixing MCMC for sampling from an EBM. We propose to learn an EBM with a flow-based backbone model. The resulting EBM in the latent space is of a simple form that is much more friendly to MCMC mixing. Our work provides strong empirical evidence regarding the feasibility of mixing MCMC sampling in EBMs parametrized by modern ConvNet for the modality of images.

## 3 MODEL AND LEARNING

### 3.1 FLOW-BASED MODEL

Let $x$ be the input example, such as an image. A flow-based model is of the form

$$z \sim q_0(z), \; x = g_\alpha(z), \tag{1}$$

where $z$ is the latent vector of the same dimensionality as $x$, and $q_0$ is a known prior distribution such as a Gaussian white noise distribution. $g_\alpha$ is a composition of a sequence of invertible transformations whose inversions and log-determinants of the Jacobians can be obtained in closed form. As a result, these transformations are of highly constrained forms. $\alpha$ denotes the model parameters. Let $q_\alpha(x)$ be the probability density at $x$ under the transformation $x = g_\alpha(z)$, then according to the change of variable,

$$q_0(z)dz = q_\alpha(x)dx, \tag{2}$$

where $dz$ and $dx$ are understood as the volumes of the infinitesimal local neighborhoods around $z$ and $x$ respectively under the mapping $x = g_\alpha(z)$. Then for a given $x$, $z = g_\alpha^{-1}(x)$, and

$$q_\alpha(x) = q_0(z)dz/dx = q_0(g_\alpha^{-1}(x))|\det(\partial g_\alpha^{-1}(x)/\partial x)|, \tag{3}$$

where the ratio between the volumes $dz/dx$ is the absolute value of the determinant of the Jacobian.

Suppose we observe training examples $(x_i, i = 1, ..., n) \sim p_{\text{data}}(x)$, where $p_{\text{data}}$ is the data distribution, which is typically highly multi-modal. We can learn $\alpha$ by MLE. For large $n$, the MLE of $\alpha$ approximately minimizes the Kullback-Leibler divergence $D_{KL}(p_{\text{data}}\|q_\alpha)$. $q_\alpha$ strives to cover most of the modes in $p_{\text{data}}$, and the learned $q_\alpha$ tends to be more dispersed than $p_{\text{data}}$. In order for $q_\alpha$ to approximate $p_{\text{data}}$ closely, it is usually necessary for $g$ to be a composition of a large number of transformations of highly constrained forms with closed-form inversions and Jacobians. The learned mapping $g_\alpha(z)$ transports the unimodal Gaussian white noise distribution to a highly multi-modal distribution $q_\alpha$ in the data space as an approximation to the data distribution $p_{\text{data}}$.

### 3.2 ENERGY-BASED MODEL

An energy-based model (EBM) is defined as follows:

$$p_\theta(x) = \frac{1}{Z(\theta)} \exp(f_\theta(x))q(x), \tag{4}$$

where $q(x)$ is a reference measure, such as a uniform distribution or a Gaussian white noise distribution as in Xie et al. (2016). $f_\theta$ is defined by a bottom-up ConvNet whose parameters are denoted by $\theta$. The normalizing constant or the partition function $Z(\theta) = \int \exp(f_\theta(x))q(x)dx = \mathbb{E}_q[\exp(f_\theta(x))]$ is typically analytically intractable.

Suppose we observe training examples $x_i \sim p_{\text{data}}$ for $i = 1, ..., n$. For large $n$, the sample average over $\{x_i\}$ approximates the expectation with respect to $p_{\text{data}}$. For notational convenience, we treat the sample average and the expectation as the same.

The log-likelihood is

$$L(\theta) = \frac{1}{n} \sum_{i=1}^{n} \log p_\theta(x_i) \doteq \mathbb{E}_{p_{\text{data}}}[\log p_\theta(x)]. \tag{5}$$

The derivative of the log-likelihood is

$$L'(\theta) = \mathbb{E}_{p_{\text{data}}}[\nabla_\theta f_\theta(x)] - \mathbb{E}_{p_\theta}[\nabla_\theta f_\theta(x)] \doteq \frac{1}{n} \sum_{i=1}^{n} \nabla_\theta f_\theta(x_i) - \frac{1}{n} \sum_{i=1}^{n} \nabla_\theta f_\theta(x_i^-), \tag{6}$$

where $x_i^- \sim p_\theta(x)$ for $i = 1, ..., n$ are synthesized examples sampled from the current model $p_\theta(x)$.

The above equation leads to the "analysis by synthesis" learning algorithm. At iteration $t$, let $\theta_t$ be the current model parameters. We generate $x_i^- \sim p_{\theta_t}(x)$ for $i = 1, ..., n$. Then we update $\theta_{t+1} = \theta_t + \eta_t L'(\theta_t)$, where $\eta_t$ is the learning rate.

To generate synthesized examples from $p_\theta$, we can use gradient-based MCMC sampling such as Langevin dynamics (Langevin, 1908) or Hamiltonian Monte Carlo (HMC) (Neal, 2011), where $\nabla_x f_\theta(x)$ can be automatically computed. Since $p_{\text{data}}$ is in general highly multi-modal, the learned $p_\theta$ or $f_\theta$ tends to be multi-modal as well. As a result, gradient-based MCMC tends to get trapped in the local modes of $f_\theta$ with little chance of mixing between the modes.

### 3.3 Energy-based model with flow-based backbone

Instead of using uniform or Gaussian white noise distribution for the reference distribution $q(x)$ in the EBM in (4), we can use a relatively simple flow-based model $q_\alpha$ as the reference model. $q_\alpha$ can be pre-trained by MLE, and serves as the backbone of the model, so that the model is of the following form

$$p_\theta(x) = \frac{1}{Z(\theta)} \exp(f_\theta(x)) q_\alpha(x), \tag{7}$$

which is almost the same as in (4) except that the reference distribution $q(x)$ is a pre-trained flow-based model $q_\alpha(x)$. The resulting model $p_\theta(x)$ is a correction or refinement of $q_\alpha$, or an exponential tilting of $q_\alpha(x)$, and $f_\theta(x)$ is a free-form ConvNet to parametrize the correction. The overall negative energy is $f_\theta(x) + \log q_\alpha(x)$.

In the latent space of $z$, let $p(z)$ be the distribution of $z$ under $p_\theta(x)$, then

$$p(z)dz = p_\theta(x)dx = \frac{1}{Z(\theta)} \exp(f_\theta(x)) q_\alpha(x)dx. \tag{8}$$

Recall equation (2), $q_\alpha(x)dx = q_0(z)dz$, we have

$$p(z) = \frac{1}{Z(\theta)} \exp(f_\theta(g_\alpha(z))) q_0(z). \tag{9}$$

$p(z)$ is an exponential tilting of the prior noise distribution $q_0(z)$. It is a very simple form that does not involve the Jacobian or inversion of $g_\alpha(z)$.

### 3.4 Learning by Hamiltonian neural transport sampling

Instead of sampling $p_\theta(x)$, we can sample $p(z)$ in equation (9). While $q_\alpha(x)$ is multi-modal, $q_0(z)$ is unimodal. Since $p_\theta(x)$ is a correction of $q_\alpha$, $p(z)$ is a correction of $p_0(z)$, and can be much less multi-modal than $p_\theta(x)$ in the data space. After sampling $z$ from $p(z)$, we can generate $x = g_\alpha(z)$.

The above MCMC sampling scheme is a special case of neutral transport MCMC proposed by Hoffman et al. (2019) for sampling from an EBM or the posterior distribution of a generative model. The basic idea is to train a flow-based model as a variational approximation to the target EBM, and sample the EBM in the latent space of the flow-based model. In our case, since $p_\theta$ is a correction of $q_\alpha$, we can simply use $q_\alpha$ directly as the approximate flow-based model in the neural transport sampler. The extra benefit is that the distribution $p(z)$ is of an even simpler form than $p_\theta(x)$, because $p(z)$ does not involve the inversion and Jacobian of $g_\alpha$. As a result, we may use a flow-based backbone model of a more free form such as one based on residual network (Behrmann et al., 2018), and we will further explore this advantage in the future work. We use HMC (Neal, 2011) to sample from $p(z)$, and push the samples forward to the data space through $g_\alpha$. We can then learn $\theta$ by MLE according to equation (6). Algorithm 1 describes the details.

---

**Algorithm 1:** Learning the correction $f_\theta$ of flow $q_\alpha$ with Neural Transport (NT-EBM).

---

**input** : Learning iterations $T$, learning rate $\eta$, batch size $m$, pre-trained parameters $\alpha$, initial parameters $\theta_0$, initial latent variables $\{z_{i,0}\}_{i=1}^m \sim q_0(z)$, observed examples $\{x_i\}_{i=1}^n$, number of MCMC steps $K$ in each learning iteration.

**output:** Parameters $\{\theta_T\}$.

**for** $t = 0 : T - 1$ **do**

    1. Update $\{z_{i,t}\}_{i=1}^m$ by HMC with target distribution $p(z)$ in equation (9) for $K$ steps.

    2. Push the $z$-space samples forward through $g_\alpha$ to obtain synthesized examples $\{x_i^-\}_{i=1}^m$.

    3. Draw observed training examples $\{x_i\}_{i=1}^m$.

    4. Update $\theta$ according to equation (6).

---

### 3.5 LEARNING BY NOISE CONTRASTIVE ESTIMATION

We may also learn the correction $f_\theta(x)$ discriminatively, as in noise contrastive estimation (NCE) (Gutmann & Hyvärinen, 2010) or introspective neural networks (INN) (Tu, 2007; Jin et al., 2017; Lazarow et al., 2017). Let $x_i^+$, $i = 1, ..., n$ be the training examples, which are treated as positive examples, and let $x_i^-$, $i = 1, ..., n^-$ be the examples generated from $q_\alpha(x)$, which are treated as negative examples. For each batch, let $\rho$ be the proportion of positive examples, and $1 - \rho$ be the proportion of negative examples. We have

$$\log \left[ \frac{P(+|x)}{P(-|x)} \right] = \log \left[ \frac{\rho}{1 - \rho} \right] - \log Z(\theta) + f_\theta(x) = b + f_\theta(x), \tag{10}$$

where $b = \log \left[ \frac{\rho}{1-\rho} \right] - \log Z(\theta)$ is treated as a separate bias parameter. Then we can estimate $b$ and $\theta$ by fitting a logistic regression on the positive and negative examples. Note, that NCE is the discriminator side of GAN. Similar to GAN, we can also improve the flow-based model based on the value function of GAN. This may further improve the NCE results.

### 3.6 GENERAL LATENT VARIABLE MODEL

The above latent space exponential tilting formulation applies to general pre-trained latent variable model $z \sim q_0(z)$, $x = g_\alpha(z)$, as long as the dimensionality of $z$ is greater than or equal to that of $x$. In the case where $z$ is of higher dimensionality than $x$, we only need to re-define $x$ to be $(x, z_0)$ where $z_0$ is a sub-vector of $z$, so that the mapping between $z$ and $(x, z_0)$ is invertible. In the case of generator network $x = g(z) + \epsilon$, we can re-define $z$ to be $(z, \epsilon)$. See Appendix A.2 for details.

## 4 EXPERIMENTS

In the subsequent empirical evaluations, we shall address the following questions:

(1) Is the mixing of HMC with neural transport, both qualitatively and quantitatively, apparent?
(2) In the latent space, does smooth interpolation remain feasible?
(3) Does the exponential tilting with correction term $f_\theta(x)$ improve the quality of synthesis?
(4) In terms of ablation, what is the effect of amount of parameters $\alpha$ for flow-based $q_\alpha$?
(5) Is discriminative learning in the form of NCE an efficient alternative learning method?

The primary concern of our work is the mixing of MCMC, which is addressed in (1) and (2). We refer to Appendix A.3 and A.4 for details on training settings and model architectures.

### 4.1 MIXING

In the following, we will recruit diagnostics to quantitatively and qualitatively address the question of mixing MCMC. We will first evaluate the famous Gelman-Rubin statistic for Markov chains running in the latent space and contrast those against chains in the data space. Then, we will evaluate auto-correlation as a weaker measure of mixing. Finally, we provide a visual inspection of Markov chains in our model and compare those with a biased model known not to be amenable to mixing.

**Gelman-Rubin.** The Gelman-Rubin statistic (Gelman et al., 1992; Brooks & Gelman, 1998) measures the convergence of Markov chains to the target distribution. It is based on the notion that if multiple chains have converged, by definition, they should appear "similar" to one another, else, one or more chains have failed to converge. Specifically, the diagnostic recruits an analysis of variance to access the difference between the between-chain and within-chain variances. We refer to the Appendix A.6 for details. Figure 2(a-b) depicts the histograms of $\hat{R}$ for $m = 64$ chains over $n = 2,000$ steps with a burn-in time of $400$ steps, learned from SVHN dataset. The mean $\hat{R}$ value is $1.13$, which we treat as approximative convergence to the target distribution (Brooks & Gelman, 1998). We contrast this result with over-damped Langevin dynamics in the latent space and HMC in the data space, both with unfavorable diagnostics of mixing.

**Auto-Correlation.** MCMC sampling leads to autocorrelated samples due to the inherent Markovian dependence structure. The $\Delta t$ (sample) auto-correlation is the correlation between samples $\Delta t$ steps apart in time. Figure 2(c-d) shows auto-correlation against increasing time lag $\Delta t$, learned

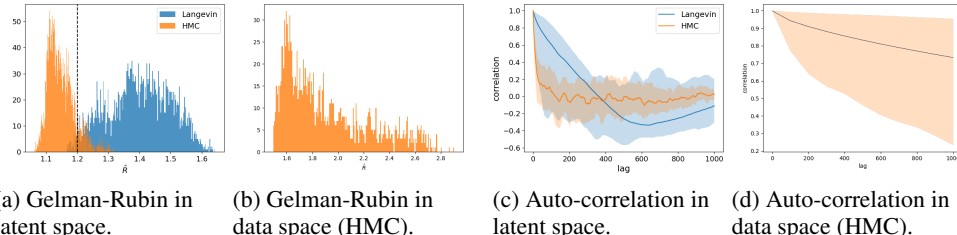

| (a) Gelman-Rubin in latent space. | (b) Gelman-Rubin in data space (HMC). | (c) Auto-correlation in latent space. | (d) Auto-correlation in data space (HMC). |

Figure 2: Diagnostics for the mixing of MCMC chains with $n = 2,000$ steps of Langevin (blue) and HMC (orange), learned from SVHN dataset. (a-b) Histograms of Gelman-Rubin statistic of multiple long-run Markov chains. $\hat{R} < 1.2$ indicates approximative convergence. (c-d) Auto-correlation of a single long-run Markov chain over time lag $\Delta t$ with mean depicted as line and min/max as bands.

from SVHN dataset. While the auto-correlation of HMC chains with neural transport vanishes within $\Delta t = 200$ steps, the over-damped Langevin sampler requires $\Delta t > 1,000$ steps, and the auto-correlation of HMC chains in the data space remains high. The single long-run Markov chain behavior is consistent with the Gelman-Rubin statistic assessing multiple chains.

**Visual Inspection.** Assume a Markov chain is run for a large numbers of steps with a Hamiltonian neural transport. Then, the Markov chains are pushed forward into data space with visualized long run trajectories in Figures 4 and 9 where $p_\theta$ is learned on the SVHN ($32 \times 32 \times 3$) (Netzer et al., 2011) and CelebA ($64 \times 64 \times 3$) (Liu et al., 2015) datasets, respectively. Figure 3 contrasts the Markov chains that sample the EBM learned with short-run MCMC (Nijkamp et al., 2019), which does not mix, against our method in which the pulled back Markov chains mix freely. We observe the Markov chains are freely traversing between local modes, which we consider a weak indication of mixing MCMC.

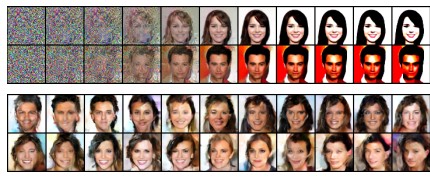

Figure 3: Long-run Markov chains for learned models without and with mixing. *Top:* Chains trapped in an over-saturated local mode. Model learned by short-run MCMC (Nijkamp et al., 2019) without mixing. *Bottom:* Chain is freely traversing local modes. Model learned by Hamiltonian neural transport with mixing.

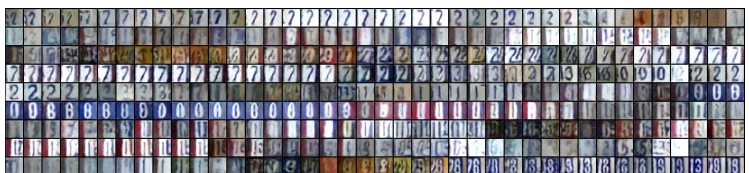

Figure 4: A single long-run Markov Chain with $n = 2,000$ steps depicted in 5 steps intervals sampled by Hamiltonian neural transport on SVHN ($32 \times 32 \times 3$).

## 4.2 INTERPOLATION

Interpolation allows us to appraise the smoothness of the latent space. In particular, two samples $z_1$ and $z_2$ are drawn from the prior distribution $q_0$. We may spherically interpolate between them in $z$-space and then push forward into data space to assess $q_\alpha$.

To evaluate the tilted model $p_\theta(z)$, we run a magnetized form of the over-damped Langevin equation for which we alter the negative energy $U(z) = f_\theta(g_\alpha(z)) + \log q_0(z)$ to $U_\gamma(z) = U(z) - \gamma \|z - z^*\|_2$ with a magnetization constant $\gamma$ (Hill et al., 2019). Note, $\frac{d}{dz}\|z\|_2 = z/\|z\|_2$, thus, the magnetization term introduces a vector field pointing with uniform strength $\gamma$ towards $z^*$. The resulting Langevin

equation is $dz(t) = \left(\Delta U(z(t)) + \gamma \frac{z(t)-z^*}{\|z(t)-z^*\|_2}\right) dt + \sqrt{2}dW(t)$ with Wiener process $W(t)$. To find a low energy path from $z_1$ towards $z_2$, we set $z^* = z_2$, $z = z_1$ and perform $n = 1,000$ steps of the discretized, magnetized Langevin equation with small $\gamma$.

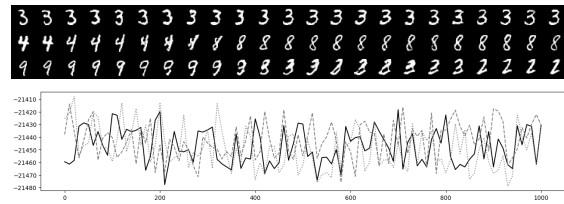

Figure 5: Low energy path between $z_1$ and $z_2$ by magnetized Langevin dynamics over $n = 1,000$ steps on MNIST ($28 \times 28 \times 1$). *Top:* Trajectory in data-space. *Bottom:* Energy profile over time.

Figure 5 depicts the low-energy path in data-space and energy $U(z)$ over time. The qualitatively smooth interpolation and narrow energy spectrum indicate that Langevin dynamics in latent space (with small magnetization) is able to traverse two arbitrary local modes, thus, substantiating our claim that the model is amenable to mixing.

## 4.3 SYNTHESIS

While the emphasis of our work is on the mixing MCMC, we do evaluate the quality of synthesis on four datasets: MNIST ($28 \times 28 \times 1$) (LeCun et al., 2010), SVHN ($32 \times 32 \times 3$) (Netzer et al., 2011), CelebA ($64 \times 64 \times 3$) (Liu et al., 2015), and, CIFAR-10 ($32 \times 32 \times 3$) (Krizhevsky et al.).

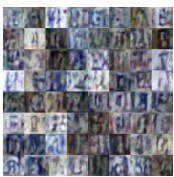 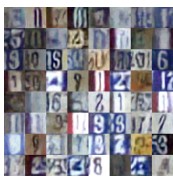 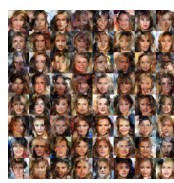 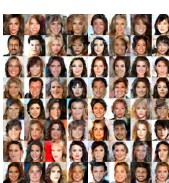

(a) Samples from flow $q_\alpha$ by ancestral sampling.    (b) Samples from $p_\theta$ by neural transport.    (c) Samples from flow $q_\alpha$ by ancestral sampling.    (d) Samples from $p_\theta$ by neural transport.

Figure 6: Comparison of generated samples by ancestral sampling from flow $q_\alpha$ and neural transport sampling from $p_\theta$ learned by NT-EBM. *Left:* SVHN ($32 \times 32 \times 3$). *Right:* CelebA ($64 \times 64 \times 3$).

The qualitative results are depicted in Figure 6 which contrast generated samples from Glow $q_\alpha$ against Markov chains by Hamiltonian neural transport from $p_\theta$. Table 1 compares the Fréchet Inception Distance (FID) (Heusel et al., 2017) with Inception v3 (Szegedy et al., 2016) on $50,000$ generated examples. Both, qualitatively and quantitatively speaking, we observe a significant improvement in quality of synthesis with exponentially tilting of the reference distribution $q_\alpha$ by the correction $f_\theta$. However, the overall quality of synthesis is relatively low in comparison to baselines (Miyato et al., 2018; Song & Ou, 2018) with FID 29.3 and 20.9 on CIFAR-10, respectively, which do not involve inference of latent variables. We hope advances in flow architectures and jointly learning the flow model may address these issues in future work.

Table 1: FID scores for generated examples in comparison to VAE (Kingma & Welling, 2013), ABP (Han et al., 2017), and Glow (Kingma & Dhariwal, 2018).

| Method | MNIST | SVHN | CelebA | CIFAR-10 |
|---|---|---|---|---|
| VAE | 32.86 | 49.72 | 48.27 | 106.37 |
| ABP | 39.12 | 48.65 | 51.92 | 114.13 |
| Glow (MLE) | 66.04 | 94.23 | 59.35 | 90.08 |
| NCE-EBM (Ours) | 36.52 | 79.84 | 51.73 | — |
| NT-EBM (Ours) | **21.32** | **48.01** | **46.38** | **78.12** |

In Table 2, we show the FID scores for samples obtained from every 1,000 steps of a single long-run chain for a model learned on SVHN. That is, the first FID score is calculated over the first

Table 2: FID scores for samples collected from a single long-run chain on SVHN ($32 \times 32 \times 3$).

| # Samples | 1,000 | 2,000 | 3,000 | 4,000 | 5,000 | 6,000 | 7,000 | 8,000 | 9,000 | 10,000 |
|---|---|---|---|---|---|---|---|---|---|---|
| FID | 97.51 | 76.74 | 67.67 | 60.41 | 60.25 | 56.45 | 55.34 | 53.85 | 52.32 | 48.72 |

1,000 consecutive samples, the second FID score over the first 2,000 consecutive samples, and so forth. The FID score converges to our reported FID score with multiple sampling chains and a fixed number of sampling steps (Table 1), which indicates that one can obtain a set of fair samples of the model by sampling from just a single very long-run HMC chain.

### 4.4 ABLATION

We investigate the influence of the number of parameters $\alpha$ of flow-based $q_\alpha$ on the quality of synthesis. In Table 3, we show at what number of parameters a small flow-based model with a small EBM correction outperforms a large flow-based model. Our method with a "medium" sized backbone significantly outperforms the "largest" Glow.

Table 3: FID scores for generated examples for $q_\alpha$ and our method with varying sizes of parameters $\alpha$ on SVHN ($32 \times 32 \times 3$). Small: $depth = 4, width = 128$, Medium: $depth = 8, width = 128$. Large: $depth = 16, width = 256$, Largest: $depth = 32, width = 512$.

| Method | Small | Medium | Large | Largest |
|---|---|---|---|---|
| Glow (MLE) | 110.55 | 94.34 | 89.31 | 86.18 |
| NT-EBM (Ours) | 74.77 | 48.01 | 43.82 | — |

### 4.5 NOISE CONTRASTIVE ESTIMATION

Noise Contrastive Estimation (NCE) is a computationally efficient learning procedure which avoids MCMC sampling by re-casting the learning problem into the form of a logistic regression. Hence, we wish to learn the correction $f_\theta$ with our NCE-EBM algorithm according to equation (10), while we still sample from the learned model with neural transport MCMC. Table 1 compares the learned models with both learning methods. The long-run Markov chains in the energy-based models learned by NCE are conducive to mixing and remain of high visual quality. Figure 11 (see Appendix A.8) depicts samples from $q_\alpha$ (left) and samples from $p_\theta$ learned by our NCE algorithm for which sampling is performed using Hamiltonian neural transport (right) for CelebA. Figure 10 (see Appendix A.7) depicts a long-run Markov chain pushed forward into data space which enjoys realistic synthesis with high diversity. This finding indicates the efficacy of learning flow backboned EBM by NCE, while, after learning the model, we may draw samples by HMC with neural transport.

## 5 CONCLUSION

This paper proposes to learn an EBM as a correction or an exponential tilting of a flow-based model, or in general a top-down latent variable model, so that neural transport MCMC sampling in the latent space of the model can mix well and traverse the modes in the data space. From a statistical perspective, the mixing of MCMC is a fundamental problem and is crucial for proper learning of EBMs. In future work, we will investigate and identify downstream tasks which significantly benefit from mixing MCMC and properly learned models by our approach.

ACKNOWLEDGEMENT

The work was supported by NSF DMS-2015577, ONR MURI project N00014-16-1-2007, DARPA XAI project N66001-17-2-4029, and XSEDE grant ASC170063. We thank Matthew D. Hoffman, Diederik P. Kingma, Alexander A. Alemi, and Will Grathwohl for helpful discussions.

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

# A APPENDIX

## A.1 CHANGE OF VARIABLE

Under the invertible transformation $x = g(z)$, let $p(z)$ be the density of $z$, and $p(x)$ be the density of $x$. Let $D_z$ be an infinitesimal neighborhood around $z$, and let $D_x$ be an infinitesimal neighborhood around $x$, so that $g$ maps $z$ to $x$, and maps $D_z$ to $D_x$. Then

$$\Pr(D_z) = \Pr(D_x). \tag{11}$$

$\Pr(D_z) = p(z)|D_z| + o(|D_z|)$, and $\Pr(D_x) = p(x)|D_x| + o(|D_x|)$, where $|D_z|$ and $|D_x|$ are the volumes of $D_z$ and $D_x$ respectively. Thus we have

$$p(z)|D_z| = p(x)|D_x|, \tag{12}$$

where we ignore $o(|D_z|)$ and $o(|D_x|)$ terms. This is the meaning of

$$p(z)dz = p(x)dx, \tag{13}$$

where $|D_x|/|D_z|$ or $dx/dz$ is the determinant of the Jacobian of $g$.

Equation (13) is a convenient starting point for deriving densities under change of variable.

## A.2 ENERGY-BASED CORRECTION AND CHANGE OF VARIABLE FOR GENERATOR MODEL

The generator model is of the form $z \sim N(0, I_d)$, and $x = g_\alpha(z) + \epsilon$, $\epsilon \sim N(0, \sigma^2 I_D)$, where $D$ is the dimensionality of $x$, and $d \ll D$ is the dimensionality of the latent vector. Unlike the flow-based model, the marginal distribution of $x$ involves intractable integral.

We shall study exponential tilting of generator model using the simple equation (13) for change of variable. To that end, we let $\tilde{z} = (z, \epsilon)$, and let $\tilde{x} = (z, x)$. Then

$$\tilde{x} = (z, x) = G_\alpha(\tilde{z}) = G_\alpha(z, \epsilon) = (z, g_\alpha(z) + \epsilon). \tag{14}$$

Let $q_0(\tilde{z})$ be the Gaussian white noise distribution of $\tilde{z}$ under the generator model. Let $q_\alpha(\tilde{x})$ be the distribution of $\tilde{x}$ under the generator model. Consider the change of variable between $\tilde{z}$ and $\tilde{x}$. In parallel to equation (13), we have

$$q_0(\tilde{z})d\tilde{z} = q_\alpha(\tilde{x})d\tilde{x}. \tag{15}$$

The marginal distribution $q_\alpha(x) = \int q_\alpha(\tilde{x})dz = \int q_\alpha(z, x)dz$, which is intractable.

Suppose we exponentially tilt $q_\alpha(\tilde{x})$ to

$$p_\theta(\tilde{x}) = \frac{1}{Z(\theta)} \exp(f_\theta(\tilde{x}))q_\alpha(\tilde{x}). \tag{16}$$

Again this can be translated into the space of $\tilde{z}$ so that under $p_\theta(\tilde{x})$,

$$p(\tilde{z})d\tilde{z} = p_\theta(\tilde{x})d\tilde{x}. \tag{17}$$

Combining equations (15), (16), and (17), we have

$$p(\tilde{z}) = \frac{1}{Z(\theta)} \exp(f_\theta(G_\alpha(\tilde{z}))q_0(\tilde{z}), \tag{18}$$

that is, under the tilted model $p_\theta(\tilde{x})$,

$$p(z, \epsilon) = \frac{1}{Z(\theta)} \exp(f_\theta(z, g_\alpha(z) + \epsilon))q_0(z, \epsilon). \tag{19}$$

We may let $f_\theta$ be $f_\theta(g_\alpha(z) + \epsilon)$, i.e., it only depends on $x = g_\alpha(z) + \epsilon$, so that it is a data space energy-based correction of the intractable $q_\alpha(x)$. In practice we may also set $\epsilon = 0$, although this is not entirely theoretically sound.

### A.3 MODEL ARCHITECTURES

For Glow model $q_\alpha$, we follow the setting of Kingma & Dhariwal (2018) with $n\_bits\_x = 8$, $flow\_permutation = 2$, $flow\_coupling = 0$.

For the EBM model $f_\theta$, we use the following Conv-Net structure.

We use the following notation. Convolutional operation $conv(n)$ with $n$ output feature maps and bias term. We recruit $LipSwish(x) = Swish(x)/1.1$ (Chen et al.) nonlinearity where $Swish(x) = x * sigmoid(x)$ (Ramachandran et al., 2017) as activation function . We set $n_f \in \{32, 64\}$.

Specifically, we set use the following hyper-parameters:

1. **MNIST**: For Glow, $n\_levels = 3$, $depth = 8$, $width = 128$. For EBM, $n_f = 32$.
2. **SVHN**: For Glow, $n\_levels = 3$, $depth = 8$, $width = 128$. For EBM, $n_f = 32$.
3. **CelebA**: For Glow, $n\_levels = 3$, $depth = 16$, $width = 256$. For EBM, $n_f = 32$.
4. **CIFAR-10**: For Glow, $n\_levels = 3$, $depth = 16$, $width = 512$. For EBM, $n_f = 32$.

| Energy-based Model ($32 \times 32 \times 3$) | | |
|:---:|:---:|:---:|
| Layers | In-Out Size | Stride |
| Input | $32 \times 32 \times 3$ | |
| $3 \times 3$ conv($n_f$), $LipSwish$ | $32 \times 32 \times n_f$ | 1 |
| $4 \times 4$ conv($2 * n_f$), $LipSwish$ | $16 \times 16 \times (2 * n_f)$ | 2 |
| $4 \times 4$ conv($4 * n_f$), $LipSwish$ | $8 \times 8 \times (4 * n_f)$ | 2 |
| $4 \times 4$ conv($4 * n_f$), $LipSwish$ | $4 \times 4 \times (4 * n_f)$ | 2 |
| $4 \times 4$ conv(1) | $1 \times 1 \times 1$ | 1 |

Table 4: Network structures for EBM with data-space ($32 \times 32 \times 3$).

| Energy-based Model ($64 \times 64 \times 3$) | | |
|:---:|:---:|:---:|
| Layers | In-Out Size | Stride |
| Input | $64 \times 64 \times 3$ | |
| $3 \times 3$ conv($n_f$), $LipSwish$ | $64 \times 64 \times n_f$ | 1 |
| $4 \times 4$ conv($2 * n_f$), $LipSwish$ | $32 \times 32 \times (2 * n_f)$ | 2 |
| $4 \times 4$ conv($4 * n_f$), $LipSwish$ | $16 \times 16 \times (4 * n_f)$ | 2 |
| $4 \times 4$ conv($8 * n_f$), $LipSwish$ | $8 \times 8 \times (8 * n_f)$ | 2 |
| $4 \times 4$ conv($8 * n_f$), $LipSwish$ | $4 \times 4 \times (8 * n_f)$ | 2 |
| $4 \times 4$ conv(1) | $1 \times 1 \times 1$ | 1 |

Table 5: Network structures for EBM with data-space ($64 \times 64 \times 3$).

### A.4 TRAINING

**Data.** The training image dataset are resized and scaled to $[-1, 1]$. We use 60,000, 70,000, 30,000, 50,000 observed examples for MNIST ($28 \times 28 \times 1$), SVHN ($32 \times 32 \times 3$), CelebA ($64 \times 64 \times 3$), and CIFAR-10 ($32 \times 32 \times 3$), respectively.

**Glow.** The parameters $\alpha$ of the flow model $q_\alpha$ are pre-trained following the configuration and reference implementation provided in (Kingma & Dhariwal, 2018). Note, the models in Table 1 and 3 have been trained based on the official reference implementations. To ensure a fair comparison

of learning Glow by MLE and our methods, we disable learning of the spatial prior and use additive coupling layers for Glow. We refer to Appendix A.3 for a specification of the Glow model configuration.

**EBM.** The network parameters are initialized with Xavier (Glorot & Bengio, 2010) and optimized using Adam (Kingma & Ba, 2015) with $(\beta_1, \beta_2) = (0.99, 0.999)$. For NT-EBM, the learning rates used are $5e-5$, $5e-5$, $1e-5$, $5e-5$ for MNIST, SVHN, CelebA, CIFAR-10, respectively and a batch-size of $64$ examples. For NCE-EBM, the learning rates used are $1e-5,1e-5,1e-5$ for MNIST, SVHN, and CelebA, respectively, and a batch-size of $128$ examples. For NT-EBM, in training the maximum number of parameter $\theta$ updates was $40,000$. For NCE-EBM, in training the maximum number of parameter $\theta$ updates was $80,000$.

**HMC.** We run Hamiltonian Monte Carlo (HMC) with persistent chains (Tieleman, 2008) initialized from $q_\alpha$ and 20 steps of MCMC and 3 leapfrog integrator steps per update of parameters of $\theta$. The initial discretization step-size $0.15$ with a simple adaptive policy multiplicatively increases or decreasing the step-size of the inner kernel based on the value of the Metropolis-Hastings acceptance rate (Andrieu & Thoms, 2008). The target acceptance-rate is set to $0.651$ (Beskos et al., 2013). Figure 7 depicts the MH acceptance-rate and adaptive step-size over time.

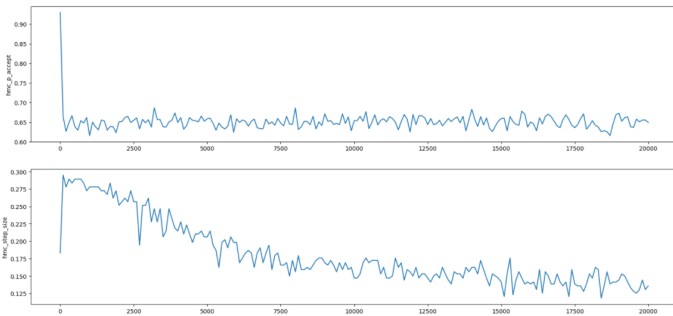

Figure 7: Metropolis-Hastings acceptance rate (top) and adaptive step-size (bottom) over time.

**FID.** The Fréchet Inception Distance (FID) (Heusel et al., 2017) with Inception v3 classifier (Szegedy et al., 2016) was computed on $50,000$ generated examples with $50,000$ observed examples as reference.

## A.5 SYNTHESIS

Figure 8 depicts samples from pre-trained flow $q_\alpha$ and samples from $p_\theta$ learned by neural transport MCMC for the dataset CIFAR-10 ($32 \times 32 \times 3$).

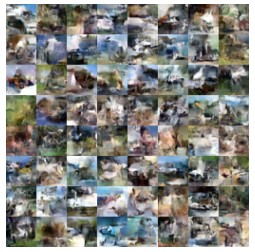
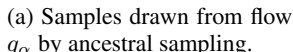
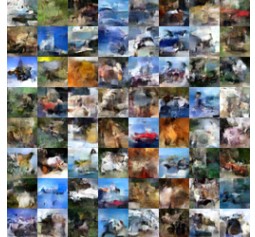

(a) Samples drawn from flow $q_\alpha$ by ancestral sampling.

(b) Samples drawn from $p_\theta$ by Hamiltonian neural transport.

Figure 8: Generated samples from a model learned by NT-EBM on CIFAR-10 ($32 \times 32 \times 3$).

### A.6 GELMAN-RUBIN STATISTIC

The Gelman-Rubin statistic (Gelman et al., 1992; Brooks & Gelman, 1998) measures the convergence of Markov chains to the target distribution. It is based on the notion that if multiple chains have converged, by definition, they should appear "similar" to one another, else, one or more chains have failed to converge. Specifically, the diagnostic recruits an analysis of variance to access the difference between the between-chain and within-chain variances.

Let $p$ denote the target distribution with mean $\mu \in \mathcal{R}$ and variance $\sigma^2 < \infty$. Gelman et al. (1992) designs two estimators of $\sigma^2$ and compares the square root of their ratio to 1. Let $X = \{X_{ij}, i = 1, \ldots, m, j = 1, \ldots, n\}$ denote $m$ Markov chains of length $n$. Let $s_w^2 = \frac{1}{m} \sum_{i=1}^m \left[ \frac{1}{n-1} \sum_{j=1}^n (X_{ij} - \bar{X}_{i\cdot})^2 \right]$ be the within-chain variance. The quantity $s_w^2$ underestimates $\sigma^2$ due to positive correlation in the Markov chain. Let $\hat{\sigma}^2 = \frac{n-1}{n} s^2 + \frac{s_b^2}{n}$ be a mixture of within-chain variance $s_w^2$ and between-chain variance $s_b^2 = \frac{n}{m-1} \sum_{j=1}^m (\bar{X}_{i\cdot} - \bar{X}_{\cdot\cdot})^2$. The quantity $\hat{\sigma}^2$ will overestimate $\sigma^2$, if an over-dispersed initial distribution for the Markov chains was used (Gelman et al., 1992). That is, $s_w^2$ underestimates while $\hat{\sigma}^2$ overestimates $\sigma^2$. Both estimators are consistent for $\sigma^2$ as $n \to \infty$ (Vats & Knudson, 2018). In light of this, the Gelman-Rubin statistic monitors convergence as the ratio $\hat{R} = \sqrt{\frac{\hat{\sigma}^2}{s^2}}$. Hence, $\hat{R}$ measures the degree to which variance (of the means) between chains exceeds what one would expect if the chains were identically distributed. If all chains converge to $p$, then as $n \to \infty$, $\hat{R} \to 1$. Before that, $\hat{R} > 1$. The heuristics $\hat{R} < 1.2$ indicates approximate convergence (Brooks & Gelman, 1998).

### A.7 VERY LONG MARKOV CHAIN

In Figure 9, the model $p_\theta$ is learned with NT-EBM on the CelebA ($64 \times 64 \times 3$) (Liu et al., 2015) dataset. In Figure 10, the model $p_\theta$ is learned with NCE-EBM on the SVHN ($3232 \times 3$) (Netzer et al., 2011) dataset.

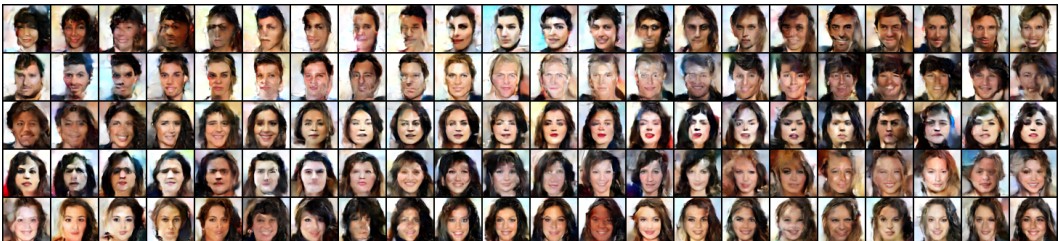

Figure 9: A single long-run Markov Chain with $n = 2,000$ steps depicted in 5 steps intervals sampled by Hamiltonian neural transport on CelebA ($64 \times 64 \times 3$).

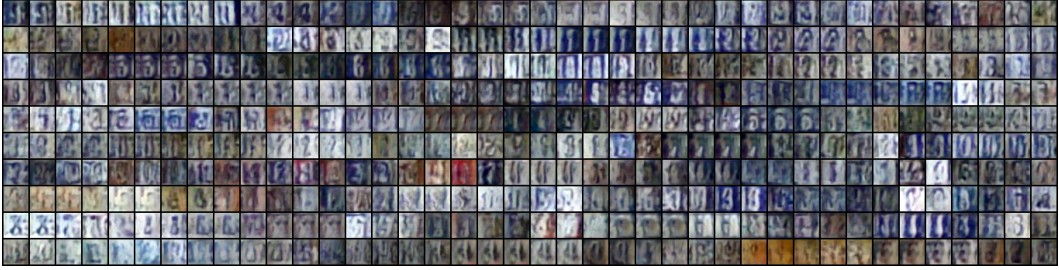

Figure 10: A single long-run Markov Chain with $n = 2,000$ steps depicted in 5 steps intervals sampled by HMC neural transport for a model learned by NCE on SVHN ($32 \times 32 \times 3$).

## A.8 NOISE CONTRASTIVE ESTIMATION

Figure 11 depicts samples from $q_\alpha$ (left) and samples from $p_\theta$ learned by our NCE algorithm for which sampling is performed using Hamiltonian neural transport (right) for CelebA.

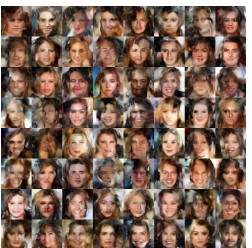       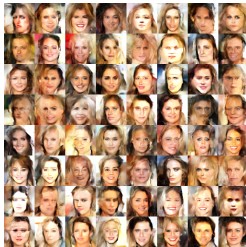

(a) Samples drawn from $q_\alpha$ by ancestral sampling.       (b) Samples drawn from $p_\theta$ by neural transport.

Figure 11: Generated samples from a model learned by NCE-EBM on CelebA ($64 \times 64 \times 3$).

