# OpenReview forum: "MCMC Should Mix: Learning Energy-Based Model with Neural Transport Latent Space MCMC"
_ICLR.cc/2022/Conference — ICLR 2022 Poster_

### Official Review · Reviewer_Kukr · 2021-10-27

**Correctness:** 3
**Technical Novelty And Significance:** 3
**Empirical Novelty And Significance:** 3
**Recommendation:** 8
**Confidence:** 2

**Main Review:**

My first suggestion is that the authors explain what the consequences of non-mixing Markov chains could be.

I was a bit worried when you said the quality of synthesis was a second issue to you. Do the authors feel that mixing is more important than good synthesis? I understand that they are related, but mixing is nice only conditioned on good synthesis.

You write on page 4, "for notation convenience". I don't see the convenience, but I see it provoking many analysts.

You should state how alpha is pretrained. Additional to this, can alpha be jointly learned with theta?

Can the authors please expand what I should conclude from Figure 2(b)?

On the same line as my second point: it seems that the authors do not think that Table 1 is the main result. If this is true, why not?

What is the cause of the missing experiments in Tables 1 and 2?

I would like the authors to tinker more about the limitations of the work. One thing I would like to know more about is the relationship of your "backbone" and what Bayesians would call a prior. Is there *some* truth in the following: are you pre-training a prior?

Minor things:

When citing multiple papers at once, consider putting them at the end of the sentence. Else, it is hard to read the sentence.

**Summary Of The Paper:**

The authors propose to learn energy-based models with a flow-based model as "backbone". This is to utilize Neural Transport and ultimately make the Markov Chains mix well in data space. This is an important problem.

**Summary Of The Review:**

I like the paper, and think it is well-written in general. The main contribution is a novel combination of existing methodologies. I am not completely familiar with the literature, but if this combination is novel I believe this to be a solid contribution. However, my evaluation is dependent on adequate responses from the authors.

---

> ### Author Response · Authors · 2021-11-20
> **Response to Reviewer**
>
> Thank you for the positive feedback and valuable suggestions. It is very encouraging that you recognize the mixing of Markov Chains as an important problem!
>
> **Q1: What are the consequences of non-mixing Markov chains?**
>
> A1: As you suggested, motivating our work by considering the consequences of (non)-mixing MCMC is a good idea. We have tried to phrase precisely this motivation in the third paragraph of the introduction (bias of the MLE estimator) with a visual empirical observation in Figure 3 (drawing “fair” samples of the model as long-run Markov chains). We will further revise the manuscript to clarify the implications of learning a model with a biased estimation of the gradient.
>
> **Q2: Is mixing more important than the quality of synthesis?**
>
> A2: We believe both aspects are important. Specifically, one could treat mixing and quality of synthesis as almost orthogonal concerns. For some downstream tasks (e.g., image generation), the quality of synthesis is of utmost importance. For other tasks (e.g., out of distribution detection), fitting the density to the data distribution properly is crucial.
>
> Regarding the quality of synthesis, relative to the baseline Glow model, the quality of synthesis is indeed significantly improved. Ideally, we feel it is fair to compare the quality of synthesis of the family of models which enjoy mixing MCMC. Further, we believe recent advances in flow-based architecture may mitigate the issue. We will conduct future experiments.
>
> Moreover, we will investigate and identify downstream tasks which significantly benefit from our approach.
>
> **Q3: Pre-training of $\alpha$ and joint learning.**
>
> A3: Regarding the pre-training of $\alpha$, the parameters of the flow model are pre-trained as described in (Kingma \& Dhariwal, 2018), for which we disabled learning of the spatial prior and use additive coupling layers for Glow (as described in our Section 4.3). As you have suggested, we revised Appendix A.4 to explicitly clarify the pre-training.
>
> Regarding joint learning, this is a good suggestion and we may learn $\alpha$ and $\theta$ jointly. The existing two-stage approach is appealing as the pre-training of the flow model with MLE is quite simple and our correction in form of the EBM can be applied to pre-trained flow models with little additional computational cost.
>
> **Q4: Conclusion of Figure 2(b).**
>
> A4: Thank you for spotting this. The intention of contrasting the two sub-plots 2 (a) and (b) is not entirely apparent, as you have suggested. We will clarify this.
>
> Under the Gelman-Rubin statistic, $\hat{R} < 1.2$ is a weak indication of convergence of long-run Markov chains. We attempt to evaluate convergence with respect to (1) running the chains in latent or data space, (2) the used sampling method of Hamiltonian or Langevin dynamics. Specifically, Figure 2(b) illustrates running the Markov chain in data space under HMC dynamics. Since mean $\hat{R} \gg 1.2$, it appears there is no indication of convergence of the Markov chains in data-space, while there might be an indication of approximate convergence for Markov chains in latent space.
>
> We will revise the x-axis to be on the same range for both sub-plots to visually clarify this.
>
> **Q5: Missing numbers in Table 2.**
>
> A5: We intended to show that even our smallest model “NT-EBM (Small)” outperforms the largest Glow model by a significant margin. With increasing size of the flow backbone model $q_\alpha$, the returns in terms of FID score are diminishing. As you have suggested, we will include the missing evaluation score for sake of completeness in the final revision. Thank you!
>
> **Q6: Prior and reference distribution.**
>
> A6: You may interpret the noise distribution $q_0$ in latent space as a prior distribution, whereas in data space $q_\alpha$ may be treated as a prior which is exponentially tilted by $f_\theta$.

---

> > ### Comment · Reviewer_Kukr · 2021-11-26
> > **Thank you**
> >
> > I thank the authors for their responses, and if they make the (small) changes as mentioned, I am happy to keep my score.

---

> > > ### Author Response · Authors · 2021-11-27
> > > **Response to Reviewer**
> > >
> > > Dear Reviewer,
> > >
> > > Thank you very much for your feedback. Yes, we will definitely revise our paper based on your insightful comments.
> > >
> > > Authors.

---

### Official Review · Reviewer_17Wo · 2021-10-28

**Correctness:** 3
**Technical Novelty And Significance:** 2
**Empirical Novelty And Significance:** 2
**Recommendation:** 6
**Confidence:** 4

**Main Review:**

Strengths:

This paper addresses the mixing problem of EBM to some extent. It is also well known that training with NCE is difficult, and it is interesting that NCE can work with a product of flow and EBM.

Questions:
1. Contribution. A key technique used in this paper is the neutral transport MCMC. The paper uses the technique for both training and sampling. The main difference from Hoffman et al. (2019) is to train with neutral transport MCMC. Thereby, it is important to verify that it is the training with neutral transport MCMC that solves the non-mixing problem instead of sampling with neutral transport MCMC. Perhaps the author can add an experiment to see what will happen if training with normal MCMC and sampling with neutral transport MCMC.

2. Since the model can generate different samples in a single MCMC chain, it can potentially improve the sampling speed. For example, to generate 100 samples, prior EBMs need to run 100 parallel MCMC, and the model in this paper only needs to run one MCMC and get samples from different timesteps. Thereby, it is worth adding FID results calculated by samples taken from one MCMC chain (using different intervals) and reporting the corresponding sampling speed up.

3. Despite the improvement of mixing, the sample quality is bad. It is worse than the MCMC trained EBM (Du & Mordatch, 2019). The author needs to give some explanation on this phenomenon.

4. Missing related work. [1*] uses a product of EBM and VAE, which has some similarity. It should be discussed in related work section.

5. Missing experimental details. The dataset used for Figure 2 is not specified. What is the y-axis of Figure 2 (a), (b)?

6. Typos. In Appendix A.6, the range of the index $i$ and $j$ is not consistent.

[1*] VAEBM: A Symbiosis between Variational Autoencoders and Energy-based Models

**Summary Of The Paper:**

This paper proposes a product of EBM and FLOW and samples in latent space to improve mixing. Such a model can be learned using both MLE or NCE. Empirically, the paper shows that a single MCMC chain of such a model can traverse through different modes.

**Summary Of The Review:**

The paper is clear and easy to understanding, but there are some questions to be addressed.

---

> ### Author Response · Authors · 2021-11-20
> **Response to Reviewer**
>
> Thank you for the detailed review and insightful feedback. Below we address specific questions.
>
> **Q1: Contribution compared to NeuTra HMC.**
>
> A1: Compared to NeuTra HMC, we propose a specific model formulation where the EBM serves as a correction of the flow model. Unlike NeuTra HMC which learns a separate flow model as the variational approximation of an unnormalized target distribution (e.g., an EBM), our resulting model takes a very simple form in the latent space which does not involve the inversion and Jacobian of the flow model. This may allow us to use more free-form flow-based models. Also for our model, the EBM does not need to reproduce the knowledge learned by the flow model, so that the EBM can take a simpler form and is easier to sample from.
>
> Your suggestion on training with MCMC in data space and sampling with NeuTra MCMC is a great point. We are investigating this setting and will include it in a later revision.
>
>
> **Q2: FID scores over a long-run MCMC chain.**
>
> A2: Following your suggestion, we have added an experiment where we run a single long-run chain of 10,000 steps, and we compute the FID score for samples obtained from every 1,000 steps for a model learned on the SVHN dataset. That is, the first FID score is calculated over the first 1,000 consecutive samples, the second FID score over the first 2,000 consecutive samples, and so forth. The discussion and results are in Section 4.1 and Appendix A.7. The FID score does converge to our reported FID score for SVHN, which indicates that indeed one can sample from a single very long run HMC Markov Chain. We will include an analysis of the potential speedup in the final revision. Thank you for this insightful comment!
>
> **Q3: Sample quality is worse than the other MCMC trained EBM.**
>
> A3: Compared to (Du \& Mordatch, 2019) which parametrizes EBMs with deep residual networks, we recruit much simpler EBMs with only a few convolutional layers, to showcase that a simple correction or exponential tilting can lead to a great improvement on the flow models (as shown in Table 2). We believe that with more advanced flow and EBM structure, the issue could be mitigated. Learning proper models with mixing MCMC and generating high-quality samples are two equally important and orthogonal concerns of training EBMs, and we focus on resolving the former concern in this paper. We shall investigate the combination of the two concerns in future work.
>
> **Q4: Related work.**
>
> Q4: Thank you for the pointer and we have added a discussion of the reference in Section 2.
>
> **Q5: Dataset of Figure 2 and label of y-Axis.**
>
> Q5: For Figure 2, the underlying model was trained on the SVHN dataset. We have revised the manuscript. Thank you for pointing this out.
>
> The y-axis of (a) and (b) in terms of a histogram indicates the frequency. We have clarified this in the caption.
>
> We have fixed the typos as you pointed out. Thank you.

---

> > ### Comment · Reviewer_17Wo · 2021-11-22
> > **Thanks for your reply**
> >
> > The experiment in Q2 is very important and I suggest to move it to the main paper. I have changed my score to 6.

---

> > > ### Author Response · Authors · 2021-11-22
> > > **Thank you!**
> > >
> > > Dear Reviewer,
> > >
> > > Once more, thank you for the positive feedback, invaluable comments, and your suggestion regarding the experiment in Q2.
> > >
> > > Indeed, as you have suggested, this experiment is quite important. We have revised the manuscript to include this experiment in Section 4.3. We will complete the experiment (including an analysis of potential "speedup") and refine the Section in the next revision.
> > >
> > > Thank you!

---

### Official Review · Reviewer_tv7r · 2021-10-31

**Correctness:** 3
**Technical Novelty And Significance:** 3
**Empirical Novelty And Significance:** 2
**Recommendation:** 6
**Confidence:** 4

**Main Review:**

-Strengths

Addressing the problem of non-mixing of MCMC for sampling from an EBM is important.
The particular class of EBM model is not new, but considering the EBM in the latent space as shown in Eq.(9) seems to be new and has the benefit that the resulting EBM Eq.(9) may be much less multi-modal than the EBM Eq.(7) in the data space.
The authors raise good questions, which are exmained by empirical evaluations.

-Weaknesses

The results are not strong. In Table 1, the FID scores over CIFAR-10 is far worse than GAN-based models [a] and EBM-based models [b]. The unusually bad FID scores hurt the quality of the paper, which makes the method, though interesting, not so convincing and useful.

Some closely related references and discussions are missed.
First, note that the EBM model in [c] is parameterized by CNNs and its MCMC sampling is mixing. Thus, the claim "To the best of our knowledge, our work is the first where MCMC sampling is mixing for EBM parametrized by modern ConvNet" in this manuscript should be revised. The idea of learning EBMs by NCE has also appeared in Wang&Ou (2018) (the reference in the paper). I suggest the authors to highlight the differences between these prior studies and this work and add relevant discussions, instead of making big claims.

[a] Takeru Miyato, Toshiki Kataoka, Masanori Koyama, and Yuichi Yoshida. Spectral normalization for generative adversarial networks. arXiv:1802.05957, 2018.

[b] Y. Song, Z. Ou. Learning Neural Random Fields with Inclusive Auxiliary Generators. arxiv 1806.00271, 2018.

[c] B. Wang and Z. Ou, “Language modeling with neural trans-dimensional random fields,” IEEE Automatic Speech Recognition and Understanding Workshop, 2017.

--update after feedback--
To acknowledge the revision and improvement of this paper, I raise the score.

**Summary Of The Paper:**

I find that this paper is a re-submission from NeurIPS2020, for which I acted as one of the reviewers. The content almost remains the same.

This paper studies the learning of a special class of EBMs, which is a correction or an exponential tilting of a flow-based model.
An interesting observation that the resulting EBM in the latent space is of a simple form that is much more friendly to MCMC mixing.
It is said that HMC sampling of the EBM in the latent space, which is a simple special case of neural transport HMC, mixes well and traverses modes in the data space. Regarding this main claim, the authors raise a number of scientific questions, which are validated by empirical evaluations.

**Summary Of The Review:**

See above.

---

> ### Author Response · Authors · 2021-11-20
> **Response to Reviewer**
>
> We will try out best to revise the manuscript according to your comments. Thank you for the insightful review and the excellent references to related work.
>
> **Q1: Quality of synthesis.**
>
> A1: We acknowledge your concern and agree that the sub-par quality of synthesis hurts the overall impression.
>
> Meanwhile, we believe both aspects, quality of synthesis and mixing behavior, are important. Specifically, one could treat mixing and quality of synthesis as almost orthogonal concerns. For some downstream tasks (e.g., image generation), the quality of synthesis is of utmost importance. For other tasks (e.g., out of distribution detection), fitting the density to the data distribution properly is crucial.
>
> Regarding the quality of synthesis, relative to the baseline Glow model, the quality of synthesis is indeed significantly improved. Ideally, we feel it is fair to compare the quality of synthesis of the family of models which enjoy mixing MCMC. Further, we believe recent advances in flow-based architecture may mitigate the issue. We will conduct future experiments.
>
> Moreover, we will investigate and identify downstream tasks which significantly benefit from our approach.
>
> **Q2: Prior art.**
>
> A2: Thank you. Both [c] and Wang \& Ou (2018) are excellent studies, which we have included in our related work. We will do our best to spread this work within our cohort.
>
> To contrast our work from these prior studies:
>
> [c] concerns the modality of languages and recruits Gibbs sampling from a discrete distribution whereas our work concerns images in continuous space for which we sample by Hamiltonian or Langevin dynamics. Moreover, we cannot find an analysis of the mixing behavior of Markov chains in [c].
>
> Wang \& Ou (2018) indeed recruits NCE to learn EBMs (as other prior art does). However, we do not claim (and have not claimed) to be the first or sole work to learn EBMs with NCE by any means. Our work demonstrates efficient learning of the model in our specific formulation by NCE. Specifically, Markov chains sampled with Neural Transport under models learned by NCE still enjoy a favorable mixing behavior, which is surprising.
>
> **Q3: Claim of the first approach to achieve mixing.**
>
> A3: We have weakened and sharpened the claim. The intention given the context of our experiments was to make the claim of ``To the best of our knowledge, our work is the first where MCMC sampling with gradient-based methods is mixing for EBM parametrized by modern ConvNet in the modality of images.''. We have revised the manuscript and believe, to the best of our knowledge, that this weakened claim is indeed true.
>
> Indeed, [c] is an excellent reference, which we have included in our discussion of related work. However, [c] concerns the modality of languages and recruits Gibbs sampling, both of which are quite different from our work. We do believe this learning algorithm may yield a well formed model, but it does not include an empirical evaluation regarding the mixing behavior of Markov chains, which is a major contribution of our work.
>
> **Q4: Claim of training EBMs by NCE.**
>
> A4: We do not claim to be the first (or only) work to train EBMs with NCE. We demonstrate the possibility of efficiently learning the model in our specific formulation (an EBM as correction or tilting of a flow model as reference distribution). And, we indicate that Markov chains under this method of learning may still enjoy the mixing behavior, see Figure 8. We will revise the manuscript to address your concern explicitly.
>
> We have cited Wang \& Ou (2018) as prior art in the original submission, as you have pointed out, and politely ask you to reconsider your argument.

---

> > ### Comment · Reviewer_tv7r · 2021-11-22
> > **Suggest to polish the main text incorporating the clarifications in the feedback and improve the presentation**
> >
> > Thanks for the response from the author(s).
> >
> > > Q1: Quality of synthesis.
> >
> > I agree with you that "both aspects, quality of synthesis and mixing behavior, are important." I have some reservation for the statement "Specifically, one could treat mixing and quality of synthesis as almost orthogonal concerns." --- this is observed for YOUR model.
> >
> > This paper cares about mixing and proposes the learning method with mixing MCMC, but the experiments are taken mainly on synthesis, which does not reflect the benefit of being mixing. This may confuse the reader, since the experiments are not taken to validate/strengthen the research hypothesis.
> >
> > "Ideally, we feel it is fair to compare the quality of synthesis of the family of models which enjoy mixing MCMC."
> >
> > If enjoying MCMC will hurt the quality of synthesis, I suggest to do experiments which will benefit from your approach, as you are also aware of.
> >
> > Instead of confusing the reader, I suggest to make clear the above issues. When the issues are not all solved in this paper, the identified and claimed future work in the feedback then should be added.
> >
> > > Q2: Prior art.
> >
> > It is good to see that the author(s) clarify the connection and difference between those prior works and this work in the feedback, which, however, are not updated in the paper.
> >
> > After the Introduction, the paper immediately claims "To the best of our knowledge, our work is the first where MCMC sampling with gradient-based methods is mixing for EBM parametrized by modern ConvNet in the modality of images.'', which, however, cannot be naturally supported/understood from reading the Introduction. The discussion about Related Work comes after. And the Related Work still not incorporate the clarifications.
> >
> > This paper has its strength. I suggest to polish the main text incorporating the clarifications in the feedback and improve the presentation. I'm happy to adjust the score if the authors take efforts to improve the paper.

---

> > > ### Author Response · Authors · 2021-11-22
> > > **Response to Reviewer**
> > >
> > > Dear Reviewer,
> > >
> > > Thank you for your insightful feedback. We agree with all your points.
> > >
> > >
> > > For Q1, following your suggestion, we have further revised both the “Introduction” (the last paragraph above Figure 1)  and “Conclusion” to address your concerns about confusing the reader:
> > >
> > > “Without proper MCMC sampling, the theory and practice of learning EBMs is on a very shaky ground. The primary goal of this paper is to address the problem of MCMC mixing, which is crucial for proper learning of EBMs. The quality of synthesis is a secondary issue in our work. We believe that fitting  EBMs properly with mixing MCMC is crucial to downstream tasks that go beyond generating high-quality samples, such as out-of-distribution detection and feature learning. We will investigate our model on those tasks in future work.”
> > >
> > > Furthermore, we are conducting experiments to explicitly demonstrate the benefits of our proposed approach.
> > >
> > >
> > > For Q2, we have moderated our claim and moved our contribution after the related work in Section 2, so that sufficient context is provided:
> > >
> > > “Our work provides strong empirical evidence regarding the feasibility of mixing MCMC sampling in EBMs parametrized by modern ConvNet for the modality of images.”
> > >
> > >  Moreover, the “Related Work” section includes similarities and differences to the prior art mentioned earlier:
> > >
> > > “Proper learning of EBMs. Wang & Ou (2017) studies the proper learning of EBMs in the modality of languages and recruits Gibbs sampling from the discrete distributions. In comparison, our work concerns images in continuous space for which we sample by gradient-based MCMC. Moreover, our work emphasizes the empirical evaluation of the mixing behavior of Markov chains.“
> > >
> > >
> > > Thank you again for your valuable comments. Please let us know if the above changes address your concerns. We will continue to try our best to revise our paper based on your suggestions.

---

> > > > ### Comment · Reviewer_tv7r · 2021-11-23
> > > > **Re: Response to Reviewer**
> > > >
> > > > Thanks for the 2nd response from the author(s).
> > > >
> > > > I see the revisions, but there are lingering concerns. The point is : When all issues cannot be solved in one paper, it is better to make explicit the issues and alert the readers, instead of confusing the readers.
> > > >
> > > > In addition to just say "The quality of synthesis is a secondary issue in our work." in the Introduction, I suggest to add results, e.g. those shown in my first review, to Table 1 and add some comments in Section 4.3 SYNTHESIS.
> > > >
> > > > btw, in the last paragraph in Page 1, regarding amortized sampling, the following two papers are in fact earlier representative work than (Grathwohl et al., 2020), which, however, made some progress.
> > > >
> > > > T. Kim and Y. Bengio. Deep directed generative models with energy based probability estimation. ICLR Workshop, 2016.
> > > > Y. Song and Z. Ou. Learning Neural Random Fields with Inclusive Auxiliary Generators. arxiv 1806.00271, 2018.

---

> > > > > ### Author Response · Authors · 2021-11-23
> > > > > **Response to Reviewer**
> > > > >
> > > > > Thank you for your prompt response! Following your advice, we have further revised our paper.
> > > > >
> > > > >
> > > > > Regarding the quality of synthesis, following your suggestion, we have further revised the introduction:
> > > > >
> > > > > “The goal of this paper is to address the problem of MCMC mixing, which is important for proper learning of EBMs. The sub-par quality of synthesis of our approach is a concern, which we believe may be addressed with recent flow architectures (Durkan et al., 2019) and jointly updating the flow model in future work.”
> > > > >
> > > > > As you have suggested, the revised Table 1 includes the [a] (SN-GAN) and [b] (Inclusive-NRF) as baselines. We will train the baselines on the remaining datasets for completeness in the final revision.
> > > > >
> > > > > The following comment in Section 4.3 further addresses the relatively low quality of synthesis:
> > > > >
> > > > > “However, the overall quality of synthesis is relatively low in comparison with baselines (Miyato
> > > > > et al., 2018; Song & Ou, 2018) which do not involve inference of latent variables. We hope advances in flow architectures and jointly learning the flow model may address these issues in future work.”
> > > > >
> > > > >
> > > > > Following your advice, the revised introduction includes both references (Kim and Bengio 2016; Song and Ou 2018). Moreover, the revised related work includes a more exhaustive list of prior art on amortized sampling:
> > > > >
> > > > > “Amortized sampling. Non-mixing MCMC sampling of an EBM is a clear call for latent variables to represent multiple modes of the original model distribution via explicit top-down mapping, so that the distribution of the latent variables is less multi-modal. Earlier works in this direction include Bengio et al. (2013); Kim & Bengio (2016); Dai et al. (2017); Song & Ou (2018); Brock et al. (2018); Xie et al. (2018); Han et al. (2019); Kumar et al. (2019b); Grathwohl et al. (2020). In this paper, we choose to use flow-based model for its simplicity, because the distribution in the data space can be translated into the distribution in the latent space by a simple change of variable, without requiring integrating out extra dimensions as in the generator model. “
> > > > >
> > > > >
> > > > > Thank you for your valuable advice that has helped us improve the clarity of our presentation. We are happy to continue to revise our paper if you have further comments.
> > > > >
> > > > > ---
> > > > >
> > > > > Durkan, Conor, et al. "Neural spline flows." Advances in Neural Information Processing Systems 32 (2019): 7511-7522.

---

### Official Review · Reviewer_Wtka · 2021-11-05

**Correctness:** 3
**Technical Novelty And Significance:** 3
**Empirical Novelty And Significance:** 3
**Recommendation:** 8
**Confidence:** 4

**Main Review:**

**General Comment**

I very much enjoyed reading this paper. The motivation is stated clearly; the problem of having a biased gradient estimation of MLE in training EBMs is certainly a reasonable problem. The proposed approach certainly makes sense and I think is a very reasonable approach to this problem. While the work is very close to [1] which may raise some concerns regarding novelty, I think it can also be seen as a strength given  that it is showing a very good application of method for posterior inference in training EBMs. Furthermore, the addition of energy-based correction is I think, while trivial, is a very neat trick that enables one to avoid computing the determinant jacobian when running HMC unlike NeuTra. The experiments I think could use some improvements, but overall demonstrate the case that the chains in the proposed approach mix better than SGLD chains. The proposed model does have a small disadvantage in the sense that it requires to train a flow model in advance. However, as the authors show in Section 4.4, even using a small flow results in a significant improvement. Overall, I am happy to recommend acceptance.

**Related Work**

I think this paper is missing some of the related work, namely [2,3]. I understand this work differs vastly from [2,3] but given that both of these papers also proposed a new method to address the problem of using MCMC sampling to compute the gradients of $\log p(x)$, I think this work should compare against them or at least discuss them within context.

**Clarity**

This paper is very well-written. There were very few ambiguities. Good job!

**Experiments**

The experiments are well-designed overall. The one I think I would have liked to see (in addition to comparison against [2,3]) is a more comprehensive ablation study of HMC vs SGLD. Figure (2) already demonstrates this but if I understand correctly, Figure (2) is run on a toy dataset. Would we have seen something similar in the case of vision datasets? In other words, how much of the gain in performance is due to using HMC instead of SGLD and how much of it is due to sampling in $z$-space instead of $x$-space? Also would have been interesting to compare against standard Contrastive Divergence where instead of initializing from noise, we initialize from a flow model for a fair comparison.

**Minor Comments**

-In the Introduction, the authors state: “Recently, Nijkamp et al. (2019) proposes to initialize short-run MCMC from a fixed noise distribution, and shows that even though the learned EBM is biased, the short-run MCMC can be considered a valid model that can generate realistic examples. This partially explains why EBM learning algorithm can synthesize high quality examples even though the MCMC does not mix.” Could you elaborate on this statement? Why does this explain that CD is able to synthesize high quality samples? Because of initialization from noise instead of data? Could you elaborate abit more on the reasoning here? Thanks


**References**

[1] Hoffman, Matthew, et al. "Neutra-lizing bad geometry in hamiltonian monte carlo using neural transport." arXiv preprint arXiv:1903.03704 (2019).

[2] Grathwohl, Will, et al. "No MCMC for me: Amortized sampling for fast and stable training of energy-based models." arXiv preprint arXiv:2010.04230 (2020).

[3] Du, Yilun, et al. "Improved contrastive divergence training of energy based models." arXiv preprint arXiv:2012.01316 (2020).



**Summary Of The Paper:**

Inspired by NueTra [1], in this paper the authors propose a new approach to train deep energy-based models (EBMs). The motivation here is that EBMs are typically trained via contrastive divergence which requires MCMC sampling in a high-dimensional space and from  a mutli-modal distribution. As a result, most MCMC sampling methods do not mix well which results in a biased estimation of MLE. To address this problem, the authors propose to train an energy-based model with a (pre-trained) backbone flow; the EBM can be interpreted as a correction step to the flow model. The flow $q_\alpha(z)$ defines a latent space $z$ with a base distribution $q_0(z)$. As pointed out in [1], we can use the flow network to additionally yield the following distribution on $z$-space: $p(z) = p(x)\frac{\delta x}{\delta z}$. Similar to NeuTra [1], the authors propose to run HMC to sample from $p(z)$ which geometrically is a simpler target, and then feed $z$ to the flow to get $x$ again. The authors perform some experiments on both toy and some of the common vision datasets to show that unlike SGLD, this approach mixes way better and is able to traverse to different local models.

**Summary Of The Review:**

I very much enjoyed reading this paper. The motivation is stated clearly; the problem of having a biased gradient estimation of MLE in training EBMs is certainly a reasonable problem. The proposed approach certainly makes sense and I think is a very reasonable approach to this problem. While the work is very close to [1] which may raise some concerns regarding novelty, I think it can also be seen as a strength given  that it is showing a very good application of method for posterior inference in training EBMs. Furthermore, the addition of energy-based correction is I think, while trivial, is a very neat trick that enables one to avoid computing the determinant jacobian when running HMC unlike NeuTra. The experiments I think could use some improvements, but overall demonstrate the case that the chains in the proposed approach mix better than SGLD chains. The proposed model does have a small disadvantage in the sense that it requires to train a flow model in advance. However, as the authors show in Section 4.4, even using a small flow results in a significant improvement. Overall, I am happy to recommend acceptance.

---

> ### Author Response · Authors · 2021-11-20
> **Response to Reviewer**
>
> Thanks for your positive evaluation and insightful feedback. Below we address specific questions.
>
> **Q1: Small disadvantage in the sense that it requires to train a flow model in advance**
>
> A1: Thank you for the insightful comment. The pre-training of the flow model by MLE is relatively simple, and our correction in the form of the EBM can be applied to the pre-trained flow model with small computational cost (i.e., small network structure and small number of sampling steps). The flow model may also be jointly learned with the EBM. We are investigating the joint training setting.
>
> **Q2: Related work**
>
> A2: Thank you for the pointers. We have included a discussion of the two related papers in the Introduction.
>
> **Q3: About Figure 2**
>
> A3: For results in Figure 2, the underlying model was trained on the SVHN dataset, not a toy dataset. We have revised the paper to clarify this point. Thank you for pointing this out. In terms of the contribution to mixing, from Figure 2 we conclude that sampling from z-space is the most important factor, and HMC is more efficient than Langevin dynamics in the sense that the auto-correlation of HMC chains vanishes within fewer sampling steps.
>
> **Q4: CD with initialization from flow**
>
> A4: Thank you for the suggestion. For results in Figure 2(b,d), we initialize the sampling in x-space from samples of the flow model as you described, where no mixing behavior exhibits. We expect that training with sampling in x-space will encounter similar mixing issues.
>
> **Q5: In the Introduction, the authors state ... elaborate a bit more on the reasoning here?**
>
> A5: Section 4 of [a] details the phenomenon of high-quality synthesis with non-mixing Markov chains. While a model learned with samples drawn after a small number of MCMC steps is a valid moment matching estimator (it matches the data distribution in terms of sufficient statistics), the learned EBM density is of much lower entropy or temperature (so that the density concentrates on a few energy minima). In simpler terms, if the EBM were learned properly (with unbiased gradient estimation of MLE), then arbitrary long Markov chains should yield "realistic" samples. It is "easy" to learn a model for which the sampling distribution under a small number of MCMC steps is moment matching the data distribution, it is "hard" to learn a model from which one can sample properly.
>
> [a] Nijkamp, Erik, et al. "Learning non-convergent non-persistent short-run MCMC toward energy-based model." NeurIPS (2019).

---

> > ### Comment · Reviewer_Wtka · 2021-11-23
> > **Respond to Authors**
> >
> > Thank you your response,
> >
> > **Q1** It would be indeed  interesting to see if it still would be stable to train the flow and $f_\theta(x)$ jointly. What I meant was more that the regarding the fact that we now require more computation and memory given that the models now consist of both a flow *and* $f_\theta$ as opposed to just $f_\theta$ (standard EBM). That said, I think this is *very* small weakness because both (1) the the trade-off is worth it and (2) the flow can still be quite small as you perfectly demonstrated in 4.4.
> >
> > **Q3** Thank you for the further clarifications. I think I may have misunderstood Figure 2(b,d). Is Figure 2.b run with SGLD or HMC? I just noticed the text says HMC but because the color was blue, I assumed it was SGLD. If it is HMC, could you consider changing the color to orange to be consistent with the legend? Thanks! Also you can consider running showing SGLD for in data-space as well.
> >
> > I had two additional questions:
> > 1. From my understating, we can obtain two sets of chains by sampling in z-space. One is the z-chains themselves and one is the x-chains obtained by feeding z-chains to the flow. And we can measure $\hat{R}$ for either of these sets of chains. In Figure 2.a, is the $\hat{R}$ computed for the chains inside the latent space? Or it is just the sampling that occurs in z-space and $\hat{R}$ is computed for x-samples?
> > 2. Why deciding $\hat{R}=1.13$ as the approximate convergence to true distribution? Shouldn't $\hat{R}=1.0$ in that case?
> >
> >
> > **Q4** Thanks! I did not find in the paper that the initialization in x-space was from the flow model and assumed that the initialization was from noise.
> >
> > **Q5** Thank you for clarification!
> >
> > **Additional Comments**
> >
> > - One thing I missed in my review is that this approach, while I believe it to be effective, requires more of a leap of faith to work compared to NueTra. In NueTra, we are minimizing a KL divergence $KL[q(x)=flow(z)||p(x)]$ which yields an equivalent minimization $KL[\mathcal{N}(z;0,I)||p(z)]$. In other words, sampling from p(z) is guaranteed to be easier than p(x) because it is being optimized to look like a standard Gaussian. In the case of this paper, we are doing MLE rather than targeting some posterior distribution. The only intuition for why it might be easier to sample in z-space is that the distribution before tilting is a Gaussian. However the tilting is now more complicated because we have $f(g(z))$ instead of $f(x)$, so I do not think it is guaranteed that sampling in z-space is necessarily going to be easier. I think the paper is still solid, but I just think author should acknowledge this if this is correct.
> > - I have read the other reviews. While I agree that the evaluation of models in terms of sample quality is important, I think researching ways of obtaining unbiased gradient estimates for MLE is also equivalently important so I still think the paper should be accepted. I also think the authors overall did an alright job of citing the related work. I was not aware of [c] (which should be cited), but as the authors pointed out, it is a model designed for NLP so I think it is distant from recent EBM literature on vision datasets.

---

> > > ### Author Response · Authors · 2021-11-24
> > > **Response to Reviewer**
> > >
> > > Thank you for your deeply insightful and inspiring comments.
> > >
> > > **Q1:** About updating the flow model.
> > >
> > > **A1:** It is indeed interesting to update both the flow model $q$ and the EBM correction together. We can do it within the joint maximum likelihood framework.
> > >
> > > Inspired by your comment on the comparison with NeuTra, we are considering the following alternative scheme:
> > >
> > > $q_{\rm new} = \arg\min_q D_{KL}(q \Vert {\rm EBM} \odot q_{\rm old})$,
> > >
> > > where the notation (${\rm EBM} \odot q$) denotes the exponential tilting of the flow model $q$ by EBM.
> > >
> > > The idea is to update $q$ to amortize the exponential tilting by EBM, i.e., we distill the EBM tilting back to $q$. At convergence, $q^*$ satisfies the fixed point condition:
> > >
> > > $q^* = \arg\min_q D_{KL}(q \Vert {\rm EBM} \odot q^*)$,
> > >
> > > i.e., if we project ${\rm EBM} \odot q^*$ back to the manifold of $q$ (by minimizing reverse KL), then we get back to $q^*$, that is, at $q^*$, the EBM tilting is "perpendicular" to the manifold of $q$. This may help to let $q^*$ to produce most of the content of the whole model, so that the EBM amounts to "minimal" correction.
> > >
> > > We will conduct experiments on joint learning, and report the results in the revision.
> > >
> > > **Q3.1:** Is Figure 2.b run with SGLD or HMC?
> > >
> > > **A3.1:** Figure 2.b depicts $\hat{R}$ for Hamiltonian Monte Carlo Markov chains with $n = 2,000$ steps in data space. You are absolutely correct. The color scheme is confusing. Following your suggestion, we will include Langevin in the data space and revise the color scheme for overall consistency.
> > >
> > > **Q3.2:** In Figure 2.a, is the $\hat{R}$ computed for the chains inside the latent space?
> > >
> > > **A3.2:** Yes, $\hat{R}$ is computed for Markov chains in latent space. We will revise and clarify this point. Thank you!
> > >
> > > **Q3.3:** Why deciding $\hat{R} = 1.13$ as the approximate convergence to true distribution?
> > >
> > > **A3.3:** If all chains converge, then as $n\rightarrow\infty$, $\hat{R}\rightarrow 1$. Before that, $\hat{R} > 1$. The heuristics $\hat{R} < 1.2$ indicates approximate convergence (Brooks & Gelman, 1998). We have revised Appendix A.6 to include this clarification. Thank you!
> > >
> > > **Q6:** Comparison with NeuTra.
> > >
> > > **A6:** Thanks for the insightful point, which we definitely agree with.
> > >
> > > The MLE of $q$ minimizes $D_{KL}(p_{\rm data} \Vert q)$ over $q$, while the MLE of EBM tilted model minimizes $D_{KL}(p_{\rm data} \Vert {\rm EBM} \odot q)$ over EBM with $q$ fixed at MLE (where again (${\rm EBM} \odot q$) denotes the EBM tilting of $q$). Although it is hoped that $q$ is close to ${\rm EBM} \odot q$ as both are approximations to the same $p_{\rm data}$, we do not do explicit optimization of $q$ as is done in NeuTra.
> > >
> > > Your comment inspires us to consider updating $q$ by minimizing $D_{KL}(q \Vert {\rm EBM} \odot q_{\rm old})$ so that the flow model $q$ absorbs most of the content of the whole model. Please see our answer A1 to your Q1 above for details. This seems aligned with the objective of NeuTra, so that latent space sampling may be more justified.
> > >
> > > Thanks again for spending your precious time on our paper and sharing your valuable insights. We will try our best to further improve our paper based on your comments.

---

### Decision · Program_Chairs · 2022-01-20

**Decision:**

Accept (Poster)

**Comment:**

The authors set up a simple combination of an energy based model and a flow based model that corrects the flow based model with an energy based term. The merits of this relative only an energy based model is improved sampling to compute the gradient. The advantage over a only flow based model is that the kinds of transforms that can be used are less limited.